# Seroconversion in Galapagos Sea Lions (*Zalophus wollebaeki*) Confirms the Presence of Canine Distemper Virus in Rookeries of San Cristóbal Island

**DOI:** 10.3390/ani13233657

**Published:** 2023-11-26

**Authors:** Julian Ruiz-Saenz, Veronica Barragan, Colón Jaime Grijalva-Rosero, Eduardo A. Diaz, Diego Páez-Rosas

**Affiliations:** 1Grupo de Investigación en Ciencias Animales—GRICA, Facultad de Medicina Veterinaria y Zootecnia, Universidad Cooperativa de Colombia, Bucaramanga 680002, Colombia; julian.ruizs@campusucc.edu.co; 2Instituto de Microbiología, Colegio de Ciencias Biológicas y Ambientales, Universidad San Francisco de Quito USFQ, Quito EC170901, Ecuador; 3Galapagos Science Center, Universidad San Francisco de Quito, Isla San Cristóbal, Islas Galápagos EC200150, Ecuador; 4Escuela de Medicina Veterinaria, Hospital de Fauna Silvestre TUERI, Universidad San Francisco de Quito USFQ, Quito EC170901, Ecuador; 5Laboratorio de Salud Animal, Instituto de Biodiversidad Tropical, Escuela de Medicina Veterinaria, Universidad San Francisco de Quito USFQ, Quito EC170901, Ecuador; 6Oficina Técnica San Cristóbal, Dirección Parque Nacional Galápagos, Isla San Cristóbal, Islas Galápagos EC200150, Ecuador

**Keywords:** conservation, Galapagos Islands, invasive species, marine mammals, morbillivirus, *Zalophus wollebaeki*

## Abstract

**Simple Summary:**

Over the last four decades, the Galapagos sea lion (GSL—*Zalophus wollebaeki*) has faced a significant population decline. An important concern is the increasing population of domestic dogs in some islands of the archipelago. These animals can be a source of various infectious diseases that can be transmitted to the GSL. An important pathogen is the canine distemper virus (CDV), causing the viral infection that generates the most concern for the agencies responsible for the management and conservation of the Galapagos pinnipeds. This virus was detected in the GSL in 2010; however, very little is known about its circulation and epidemiology. Our study tested 110 GSL serum samples that were collected during the summer of 2016 and 2017. Our results showed an increasing circulation of CDV and highlight the importance of monitoring emerging diseases that can be transmitted from the domestic to the wildlife species of the archipelago.

**Abstract:**

Background: The emblematic Galapagos sea lion (GSL—*Zalophus wollebaeki*) has faced an important population decline over the last four decades. There are multiple environmental and biological factors that might be implied in this decrease. Recently, evidence of various zoonotic infectious diseases that can be potential threats has been reported. Considering that in some islands of the archipelago the risk of transmission of infectious diseases may be promoted by the increasing population of domestic dogs, epidemiological vigilance and search of new pathogens are essential. The canine distemper virus (CDV), one of the viral pathogens that generate the most concern for the agencies responsible for the management and conservation of the Galapagos pinnipeds, was detected in the GSL in 2010. However, there is scarce information about its impact on GSL health and about its epidemiology. Methods: In this study, 110 GSL serum samples were collected during the summer of 2016 and 2017. All samples were exposed to VERO dog SLAM cells expressing the canine SLAM receptor. Results: Our results showed a significative increase (*p* = 0.04) in the frequency of neutralizing antibodies to CDV in the 2017 (53.1%) samples compared to the 2016 samples (19.6%). Conclusions: Our work confirmed the continuous and increasing circulation of the CDV in the GSL and highlights the importance of monitoring emerging diseases that can be transmitted from domestic to wildlife species. Vigilance of CDV is essential to understand the role of this virus in GSL mortality and to take informed decisions for wildlife conservation.

## 1. Introduction

The endemic Galapagos sea lion (GSL, *Zalophus wollebaeki*) has been listed as an endangered species in the International Union for Conservation of Nature Red List since 2008 due to a strong reduction in its population of more than 50% over the last four decades [1]. The main factors causing this decline are environmental variations and oceanographic warming events such as the El Niño–Southern Oscillation [2,3]; however, habitat degradation and pathogen transmissions by introduced and domestic species (i.e., rats, dogs and cats) have been identified as a potential conservation problem for this species [4,5].

The permanent human population of the Galapagos archipelago has increased exponentially from 5000 people in 1974 to over 25,000 people in 2015 [6,7]. This has generated a domestic canine overpopulation on several populated islands of the archipelago [8,9]. The last assessment reported that the human/dog ratio was 4.04:1, representing an increase of 55% in the dog population on Santa Cruz Island from 2014 to 2018 [8,10]. This growing canine population can increase the risk of transmission of infectious diseases, such as that caused by the canine distemper virus (CDV), which could severely affect endangered species such as the GSL [11,12]. In turn, parasitic diseases such as heartworm (*Dirofilaria immitis*) disease and bacterial infections caused by *Leptospira* spp. or *Mycoplasma* spp. have also been reported in GSLs [11,13,14], associating all these infections with a zoonotic potential that promotes their transmission from this endemic species to the domestic fauna of the archipelago.

CDV, a paramyxovirus of the genus *Morbillivirus*, is one of the viral pathogens that generate the most concern for the agencies responsible for the management and conservation of the Galapagos pinnipeds [4]. During a high GSL pup mortality event in an El Malecón rookery on San Cristóbal island, CDV RNA was detected in tissue samples (i.e., lung and placenta) from six individuals; four of these samples were analyzed by nucleotide sequencing, showing 99% similarity to CDV [11]. Since CDV is a multi-host pathogen [15] commonly causing a viral infectious disease highly prevalent in domestic dogs and other carnivores, it may poses a conservation threat to several endangered species around the world [16,17], including South America, where the circulation of at least four different lineages of CDV was confirmed [18,19], with the South/NorthAmerica-4 lineage present in Ecuador [20].

CDV infection in pinnipeds was associated with CDV transmission from terrestrial carnivores at the aquatic–terrestrial interface through respiratory exudates, excretions and secretions containing the virus [16,21]. In addition, the GSL gregarious conditions and social behavior are likely to contribute to the rapid spread of the virus within their populations in the wild [22]. Despite the genetic confirmation of CDV, the source of exposure for the detected GSL cases has not been confirmed so far [11]. However, multiple reports of CDV circulation in dogs in the Galapagos Islands were published, indicating that the serological prevalence increased from 22% in 2004 [23] to 36% in 2014 [10], with well-documented outbreaks of CDV killing more than 600 dogs on Santa Cruz and Isabela islands in the early 2000s [10,23]. Recently, a study conducted on Santa Cruz Island reported a total of 125 dogs with clinical signs compatible with CDV infection, with 74.4% (IC95%, 66–81%) of the animals found positive by RT-qPCR for CDV detection [12]. This motivated vaccination campaigns for domestic dogs in the archipelago [12]; however, virus spread at dog–wildlife interfaces is generally not controlled [24]. Hence, the recent vaccination policies would not prevent CDV infections, which remain a threat for the endangered GSL [12].

Despite strong evidence of the threat of CDV spread in the Galapagos Islands, data related to the circulation of this disease in endemic fauna are scarce. Here, we present the first study evaluating the seropositivity to CDV in GSLs, providing information that can improve monitoring and conservation strategies for this endangered species.

## 2. Materials and Methods

### 2.1. Ethics Statement

This research was approved by the Galapagos National Park Directorate (GNPD) under the research permits PC-16-16, PC-16-17, PC-36-21 and PC-19-22 and by the Ministry of the Environment of Ecuador under the Framework Contract for Access to Genetic Resources MAATE-DBI-CM-2021-0178 granted to Dr. Diego Páez-Rosas from the University San Francisco de Quito (USFQ).

The methods described here were reviewed and approved by the GNPD and USFQ committees responsible for assessing animal welfare in research activities on the Galapagos Islands. All animals sampled in this study were wild and not under human care. They were monitored by a veterinarian, researchers, and GNPD rangers during the study.

### 2.2. Sample Collection

Blood samples were collected from juvenile animals from the El Malecón rookery on San Cristóbal Island during the summer of 2016 (n = 46) and 2017 (n = 64). Juveniles are a representative sample of GSL populations, since they are the second most abundant sex/age category in rookeries [3,25]. In addition, the capture of juveniles represents a minor disturbance to wild animal rookeries. The El Malecón rookery hosts the largest GLS population in the archipelago [4,5,25] and is located within the urban limits of Puerto Baquerizo Moreno, where dogs are frequently observed either loose or with their owners [4,5,25].

The GSLs were captured by placing a net in front of them and encouraging them into the net. At the time of capture, the individuals were weighed using an electronic scale; subsequently, each animal was removed from the net and manually held by experts in handling this species. It is important to note that each captured animal was marked with a numbered plastic tag, preventing the possibility of resampling the same individual. A physical examination was performed by a veterinarian, and morphometric measurements were taken for further studies. Blood samples were obtained by venipuncture of the caudal gluteal vein and placed in sterile vacuum tubes with ethylenediaminetetraacetic acid (EDTA) as an anticoagulant. Blood was stored in a cooler and processed within the next 6 h at the Galapagos Science Center facilities. All samples were centrifuged for 15 min at 3000 G, and the plasma was stored at −20 °C in separate tubes.

### 2.3. Laboratory and Data Analysis

The assays were performed in VERO dog SLAM cells expressing the canine SLAM receptor. The cells were cultured in DMEM (Dulbecco’s Modified Eagle Medium, GIBCO^®^, Grand Island, NY, USA) supplemented with 2% fetal bovine serum (FBS, GIBCO^®^) and a 1% antibiotic/antimycotic solution (streptomycin, 10 mg/mL, 10,000 U/mL of penicillin and 0.025 mg/mL of amphotericin B, GIBCO^®^) and incubated in a humid atmosphere with 5% CO2 at 37 °C. A clinical isolate of CDV with a titer of 2.8 × 10^8^ PFU was used in all experiments.

The serum samples from the GSLs, were examined for the presence of antibodies to CDV by the virus neutralization test according to the classic method [26]. Briefly, 30,000 VERO dog SLAM cells per well were seeded in 96-well plates. After 24 h, two-fold dilutions of each serum sample were prepared with serum-free DMEM, starting from a 1:2 dilution up to a 1:128 one. Then, 100 TCID50 of CDV were added to each dilution, followed by incubation at 37 °C for 1 h in 5% CO2. The mixture was incubated for 1 h at 37 °C before inoculation into the cells for 2 h. Then, the medium was removed, the cells were washed with PBS, and a semi-solid medium with 0.3% agarose was added. The plates were incubated for 120 h, after which time the cells were fixed with 4% formaldehyde and stained with crystal violet. Next, the antibody titer of each serum sample was established as the reciprocal of the last dilution of serum to neutralize 100 TCID50 in 50% of the wells. Neutralizing titers equal to or less than 1:8 were considered negative. A vaccinated canine serum was used as a positive control and an equine serum as a negative control.

All data were stored into an Excel^®^ database. Descriptive analysis was performed for all variables. The virus prevalence was analyzed with the WinEpi on-line package (available at http://www.winepi.net accessed on 10 August 2023), considering the total sea lion population size of 2188 individuals estimated in 2015 [25].The values are presented as absolute values with 95% confidence intervals (95% CIs). Two-way ANOVA was used to evaluate the difference in neutralization between years, and the chi-square test was used to compare the frequencies of the neutralization variables. The data were analyzed using GraphPad Prims™ v7.05 software for Windows^®^.

## 3. Results

A total of 110 serum samples from GSLs were analyzed. CDV prevalence for 2016 was 19.6% (95% CI: 8.3–30.9%), while for 2017, it was 53.1% (95% CI: 41.2–65.1%). A significative increase in the frequency of neutralizing antibodies to CDV was observed in the 2017 samples compared to the 2016 samples (*p* = 0.04), even with much higher serum neutralizing titers (≥1:128), as can be seen in Figure 1. Also, the proportion of negative GSL serum samples was significantly lower in 2017 in comparison to 2016 (*p* < 0.01), indicating an increasing CDV exposure between 2016 and 2017.

## 4. Discussion

The GSL is one of the most emblematic species of the Galapagos archipelago. Its relevance is based on ecological considerations [27], since it plays a significant role in maintaining the functional biodiversity of the Galapagos marine ecosystem [28]. Thus, there is a need to increase research efforts to understand the role of infectious diseases in GSL populations and to facilitate the early detection of emerging diseases that could further threaten this endangered pinniped [4].

Our work confirmed the ongoing circulation of CDV in Galapagos wildlife. This, combined with the increasing population of domestic dogs on the islands [8,10], represents an increased risk of infection for GSLs. We recognize that one of the limitations of this report is the lack of samples from dogs in the vicinity of the sampled rookeries; however, reports from previous years demonstrate the continuous circulation of this virus in dogs that inhabit several islands of the archipelago, including San Cristóbal island [10,12,23]. The CDV has been widely recognized as an important pathogen putting at risk the conservation of the endangered mammals [24,29]. CDV outbreaks were detected previously in other wildlife pinnipeds; for example, in 1988 and 2000, this virus was identified as the cause of death of thousand Baikal seals (*Phoca sibirica*) [30] and more than 10,000 Caspian seals (*Phoca caspica*) [31]. In addition, the mortality of the crabeater seal *(Lobodon carcinophaga)* in 1955 was retrospectively associated with the transmission of CDV from sled dogs in Antarctica [32].

In most of the cases, amplicon-based sequencing allowed the confirmation of CDV infection [31,33]; however, a strong phylogenetic support is still missing to fully understand and propose possible mutations associated with the interspecies transmission of the virus [15]. The same is true for the GSL cases detected in 2011, for which the comparison of the 149 bp nucleotide sequences from the four positive samples with sequences in GenBank showed 99% similarity to CDV [11]; however, no phylogenetic inference to the origin or possible ancestors could be indicated.

Although most conservation strategies for the GSL have focused on the control of the interactions with the domestic fauna (i.e., dogs and cats) [4], our results suggest that CDV infection is stably transmitted between sea lions or between dogs and sea lions, even in the absence of clinical signs, as has been proposed for terrestrial wildlife [24], with subclinical or clinically recovered shedders being a potential source of virus for naïve animals [34]. However, the clinical signs associated with CDV infection in wildlife vary greatly among species and depend on several factors, such as CDV strain virulence and host age and its immune status [21]. Therefore, the initial signs of CDV infection can be subtle and rarely observed [35]. Even in domestic dogs, 50–70% of CDV infections are thought to be subclinical [36].

These conditions highlight the importance of understanding the dynamics of CDV transmission in GSLs to determine if the high prevalence of anti-CDV antibodies is maintained over time, establish the consequences of CDV infection and identify the CDV strain circulating in this region. It is also important to determine the role of domestic dogs in the eco-epidemiology of CDV circulating in the Galapagos archipelago and whether GSLs could spread this virus to other marine mammals in other regions, given the sporadic occurrence of this species along the coasts of the Tropical Eastern Pacific [37,38]. Therefore, there is a need for an updated evaluation of the risk of CDV transmission and infection in GSL populations to implement management actions, such as possible vaccination campaigns. Although vaccination in free-living marine mammals is challenging [39,40], pinniped vaccination against CDV is justified when endangered species are at risk, as occurred for the Mediterranean monk seal (*Monachus monachus*) [39] or the Hawaiian monk seal (*Monachus schauinslandi*) [41]. However, vaccination using common dog vaccines (attenuated, live CDV vaccines) is contraindicated due to the potential risk of disease in the vaccinees and transmission to in-contact animals [41].

A commercially available attenuated CDV vaccine was successfully used to immunize harbor (*Phoca vitulina*) and grey seals (*Halichoerus grypus*) during the 1988 PDV epidemic [42]. Recently, a recombinant canarypox-vectored CDV vaccine was proved to be safe and effective in different captive marine mammals, including harbor seal, Hawaiian monk seal, southern sea otters (*Enhydra lutra nereis*), Steller sea lion (*Eumetopias jubatus*) and walruses (*Odobenus rosmarus*) [43,44,45]. Currently, multiple possibilities are been analyzed to establish the optimal CDV vaccination strategy for the free-ranging Hawaiian monk seal, including the administration of vaccination only in response to outbreaks, the use of prophylactic vaccination or a combination of these two approaches [46]. In addition, the implementation of a SEIR (susceptible, exposed, infectious, recovered) compartmentalized model to simulate the virus trajectories and prevent CDV outbreaks is being considered [47,48].

The increasing frequency of CDV seropositivity in GSLs highlights the need to increase management and conservation efforts for this species by the Galapagos National Park Directorate and the Ministry of the Environment of Ecuador. Hence, measures should be taken to reduce the risk of this promiscuous viral pathogen in Galapagos pinnipeds, such as: (1) routine surveillance of dogs for the detection of CDV, (2) strict control and treatment of positive dogs, (3) implementation of a strategic control program with administration of vaccines against CDV to all dogs on populated islands, (4) assessment of the viability and the effects of a potential vaccination against CDV in the Galapagos pinnipeds through continuous monitoring programs of the wild fauna of the archipelago, (5) health monitoring of the GSL populations to identify the effects of CDV or other pathogens to which this species may be exposed.

Scientific and economic efforts must also be strengthened not only to control CDV transmission from the domestic to the endemic fauna [4,5], but also to understand the potential establishment of asymptomatic/subclinical CDV infections in marine wildlife populations. To date, research efforts have predominantly focused on assessing marine mammal exposure to CDV through serology and disease diagnostics [22]. A better understanding of the complexity of CDV transmission will depend on the use of molecular techniques that facilitate epidemiological investigations and determine whether sustained amplification of a morbillivirus in a pinniped population leads to adaptive mutations in the viral genome [22].

## 5. Conclusions

Our research confirmed the circulation of the subclinical/asymptomatic CDV infection in the endangered Galapagos sea lions. Further research is needed to understand the role of CDV in disease and its interspecies transmission between domestic and marine fauna based on pathological, virological and molecular analyses.

## Figures and Tables

**Figure 1 animals-13-03657-f001:**
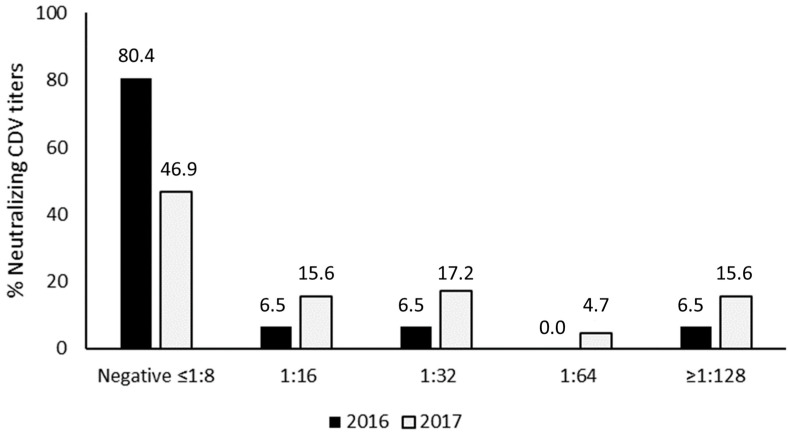
CDV-neutralizing titer in *Zalophus wollebaeki* during 2016–2017.

## Data Availability

The original contributions presented in the study are included in the article; further inquiries can be directed to the corresponding authors.

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
