# Peer review of "Seroconversion in Galapagos Sea Lions (Zalophus wollebaeki) Confirms the Presence of Canine Distemper Virus in Rookeries of San Cristóbal Island"

_animals, 2023, doi:10.3390/ani13233657_

Round 1

Reviewer 1 Report

Comments and Suggestions for Authors

Dear Authors,

congratulations on your excellent work.

My only suggestion is to include a more extended consideration of your perspective and proposal on conservation. The results of your work and the efforts behind should be used to design conservation management and policy strategies.

Many thanks

Author Response

Dear Reviewer,

I am writing you on behalf of all authors. We thank you for appreciating our work and for your valuable comment. We have extended incorporated in our discussion on conservation and policy strategies. 

Please find the editions in the attached document.

Best regards,

Verónica

Reviewer 2 Report

Comments and Suggestions for Authors

Dear Author,

The manuscript is very interesting and needs minor revisions. Please check the notes in the manuscript.

Thank you,

Best regard,

Reviewer

Comments on the Quality of English Language

The manuscript is well written

Author Response

Dear reviewer,

I am writing you on behalf of all authors. We thank you and the reviewers for appreciating our work and for all your valuable comments. We have considered all your suggestions, elaborated responses to your questions, and incorporated changes to address your concerns.  

Please find below the point-by-point responses.

Best regards,

Verónica Barragán

Reviewer 2

Comment 1: Is there any possibility that sealions were recapture?

Thanks for your comment. There is no possibility of sampling an animal twice as all sealions are marked with a numbered plastic tag on their flipper. These tags are used to monitor the sealion population.

Comment 2. Contact of Sealions with wild dogs.

There are no feral dogs on the Islands, contact of these animals with the sea lions occurs on the beach boardwalk. It is not uncommon to see dogs off leash on the beach boardwalk, where sea lions come to rest even on the benches.

Comment 3. Clinical signs in sampled sealions.

No, all animals were apparently healthy.

Reviewer 3 Report

Comments and Suggestions for Authors

In reference to the work titled "Seroconversion in Galapagos sea lions (Zalophus wollebaeki) confirms the presence of Canine Distemper Virus in rookeries of San Cristóbal Island" which describes the possible increase in cases of GSL infected with CDV, I would like to comment that it is an interesting work that shows how potentially a disease typical of one animal species can be transmitted to another that would seem to us very distant.

However, there are some doubts and points in this regard that can improve the MS.

Line 121. Why only juveniles? explain in the discussion section.

Line 122. Was there no other more recent data, for example within the last 5 years to now?

Line 123. Were any stray dogs or dogs from the area surrounding the GSL capture area analyzed? Describe.

Line 184. In fact, if comparing between 2016 and 2017, this could be a trend or just a chance to increase in 2017. How would you prove that there is indeed a real increase in the last 5 years without having sampled dogs or GSL? Explain.

Line 185. Parallel sampling of the dogs would be required. Why is there not even a sample of feral or domestic dogs in the area of sampling to compare if CDV is in the zone? Explain

Line 188. What would be these new efforts and by whom?

Line 205. Although this possibility has been shown, dogs were not sampled in this study. Explain why

Line 220. Why think only about a vaccination campaign, it could also be a campaign to determine if dogs from the Galapagos or the region close to GSL have a high prevalence of CDV. Explain

Line 227. In addition to these measures, what other types of actions would you promote? Explain

Comments on the Quality of English Language

The MS generally has a good English structure, it only requires small details that must be attended to.

Author Response

Reviewer 3

Dear reviewer,

I am writing you on behalf of all authors. We thank you for appreciating our work and for all your valuable comments. We have considered all your suggestions, elaborated responses to your questions, and incorporated changes to address your concerns.  

Please find below the point-by-point responses.

Best regards,

Verónica Barragán

Line 121. Why only juveniles? Explain in the discussion section.

Thank you for bringing this to our attention. Juveniles are the second most abundant sex and age group in sea lion colonies and were captured for sampling because this procedure causes less disturbance to the colonies. The most abundant sex and age group are females, but we didn't capture this group because of the high probability of finding pregnant adult female sea lions.

We have added “Juveniles are a representative sample of the sea lion population as they are the second most abundant population in the colony. In addition, the capture of these animals represents a minor disturbance to the colony.” to line 123.

Line 122. Was there no other more recent data, for example within the last 5 years to now?

Unfortunately, the logistics of sampling Galapagos sea lions are very complicated and resource intensive. In addition, from 2020 to 2022, pandemic restrictions prevented entry to the islands. We believe it is very important to continue monitoring CDV in sea lion populations, and publishing the results of our work will be a great help in raising more funds to continue the research.

Line 123. Were any stray dogs or dogs from the area surrounding the GSL capture area analyzed? Describe.

We were not able to analyze dog samples, however previous reports have shown high CDV positivity in these animals from the archipelago, confirming the circulation of the virus. Additionally, the presence of domestic dogs, either loose or with their owners in this area, is a risk factor for exposure to diseases that can be potentially transmitted from dogs to sealions. We have added the following phrase: “… where dogs are frequently observed either loose or with their owners.” (line 127)

Line 184. In fact, if comparing between 2016 and 2017, this could be a trend or just a chance to increase in 2017. How would you prove that there is indeed a real increase in the last 5 years without having sampled dogs or GSL? Explain.  Line 185. Parallel sampling of the dogs would be required. Why is there not even a sample of feral or domestic dogs in the area of sampling to compare if CDV is in the zone? Explain.  Line 205. Although this possibility has been shown, dogs were not sampled in this study. Explain why

Thanks for your relevant comments, we are aware of the limitation of not having samples from dogs during the period of the study and the following years.

It is important to mention that CDV seropositivity in dogs from islands in the archipelago has been described, and the most recent was an outbreak was reported on 2019 in Santa Cruz Island. Additionaly, on 2011 seropositivity was detected in dogs from San Cristobal Island, and on 2009-2012 genetic material of CDV was reported in only one sealion individual. Our work indeed confirms the circulation of CDV in the Galapagos wildlife. To be more precise, we have edited line 184: “Following the detection of CDV in six sea lions reported in 2011, our work confirms the ongoing circulation of CDV in Galapagos wildlife. This, combined with the increasing population of domestic dogs on the islands, may also indicate an increased risk of infection in sea lions. We recognize that one of the limitations of this report is the lack of samples from dogs on the island or in the vicinity of the sampled colony. However, reports from previous years on the islands of San Cristobal, Santa Cruz and Isabela, as well as the outbreak reported in 2019, demonstrate the continuous circulation of this pathogen in dogs.”

Line 188. What would be these new efforts and by whom?

Thank you for noticing that we were missing this information. We have added the following to the text: “Therefore, the circulation and increasing frequency of CDV seropositivity in the GSL highlights the need for renewed efforts by the Galapagos Islands authorities and the community to reduce the risk of this promiscuous viral pathogen in Galapagos marine mammals.”

Line 220. Why think only about a vaccination campaign, it could also be a campaign to determine if dogs from the Galapagos or the region close to GSL have a high prevalence of CDV. Explain

Thank you for your comment, there is indeed a government institution which, among other things, is responsible for monitoring domestic animal diseases that could threaten the islands' wildlife. Unfortunately, these institutions do not always function as we would hope; even in recent months, the political situation has made it difficult to get access to some animal samples for infectious disease testing. However, our research team will be analyzing samples from dogs and sea lions on San Cristobal Island in a few months' time. This will give us data on the status of CDV in dogs and GSL of the island.

 We have edited the following phrase:

“Therefore, there is a need for an updated evaluation of the CDV risk of transmission and disease on GSL populations to implement management actions, such as CDV prevention in dogs and possible vaccination campaigns in GSL.”

Line 227. In addition to these measures, what other types of actions would you promote? Explain

We believe that the edits made for the above comments explain this concern.

Reviewer 4 Report

Comments and Suggestions for Authors

This ms represents a very solid and stron piece of science that would reach a high number of readers and will give adequate instruments to mitigate and resolve zoonotic problems for the Galapagos Sea lion. I wish to congratulate the authors on their excelent work.

Comments on the Quality of English Language

Minor edits and overall review is needed.

Author Response

Dear reviewer,

I am writing you on behalf of all authors. We thank you for appreciating our work. Please find attached the final version of the manuscript.

Best regards,

Verónica Barragán

Round 2

Reviewer 2 Report

Comments and Suggestions for Authors

Dear Author,

Thank you for revising the manuscript.

Best Regard,

Reviewer 3 Report

Comments and Suggestions for Authors

No further questions

Comments on the Quality of English Language

No additional comments.